# Connecting Hippo Pathway and Cytoophidia in *Drosophila* Posterior Follicle Cells

**DOI:** 10.3390/ijms25031453

**Published:** 2024-01-25

**Authors:** Rui-Yu Weng, Lei Zhang, Ji-Long Liu

**Affiliations:** 1School of Life Science and Technology, ShanghaiTech University, Shanghai 201210, China; wengry@shanghaitech.edu.cn (R.-Y.W.);; 2State Key Laboratory of Cell Biology, Center for Excellence in Molecular Cell Science, Shanghai Institute of Biochemistry and Cell Biology, Chinese Academy of Sciences, University of Chinese Academy of Sciences, Shanghai 200031, China; 3Sheng Yushou Center of Cell Biology and Immunology, School of Life Sciences and Biotechnology, Shanghai Jiao Tong University, Shanghai 200240, China

**Keywords:** CTP synthase, cytoophidium, *Drosophila melanogaster*, Hippo pathway, posterior follicle cells

## Abstract

CTP synthase (CTPS), the rate-limiting enzyme in the de novo synthesis of CTP, assembles into a filamentous structure termed the cytoophidium. The Hippo pathway regulates cell proliferation and apoptosis. The relationship of the nucleotide metabolism with the Hippo pathway is little known. Here, we study the impact of the Hippo pathway on the cytoophidium in *Drosophila melanogaster* posterior follicle cells (PFCs). We find that the inactivation of the Hippo pathway correlates with reduced cytoophidium length and number within PFCs. During the overexpression of CTPS, the presence of Hippo mutations also reduces the length of cytoophidia in PFCs. In addition, we observe that knocking down CTPS mitigates *hpo* (Hippo)-associated over-proliferation. In summary, our results suggest that there is a connection between the Hippo pathway and the nucleotide biosynthesis enzyme CTPS in PFCs.

## 1. Introduction

The nucleotide metabolism plays a crucial role in various cellular processes, including the synthesis of DNA and RNA and the generation of cellular energy. The synthesis of purines and pyrimidines follows two primary pathways: the de novo nucleotide synthesis and the salvage pathway. CTP synthase (CTPS) assumes the role of the rate-limiting enzyme in the de novo synthesis pathway for CTP [1,2]. CTPS exhibits a dynamic equilibrium among monomers, dimers and tetramers [3]. In the presence of a substrate, CTPS presents at a high concentration of the active tetramer type [4]. The activity of CTPS is increased in cancer cells, such as those in liver neoplasia and kidney tumors [5].

In 2010, CTPS was discovered to form filamentous structures, known as cytoophidia [6]. Cytoophidia are highly conserved in various species, including *Saccharomyces cerevisiae* [7], *Schizosaccharomyces pombe* [8] and mammalian cells [9]. Furthermore, the formation of cytoophidia is regulated by *Myc* [10], *Ras* [11], the mTOR-S6K1 pathway [12,13] and nutritional conditions [14] in the *Drosophila melanogaster*. Cytoophidia also appear in cancer cells, such as those found in human hepatocellular carcinoma [15].

Cytoophidia are widely distributed during both the larval stage [16] and oogenesis of *Drosophila*. Within the *Drosophila* ovaries, cytoophidia are observed in various cell types, including follicle cells, nurse cells and oocytes [17,18,19]. Cytoophidia can be categorized into two distinct classes: micro-cytoophidia, measuring approximately 1–4 μm in length, and macro-cytoophidia, which can extend to up to 30–40 μm [6]. Functionally, cytoophidia serve as scaffolds [20,21,22] and effectively prolong the half-life of CTPS [23]. Considering that CTPS is a rate-limiting enzyme, it is crucial to gain a comprehensive understanding of its regulatory mechanism.

The Hippo pathway was first discovered in *Drosophila* in 1995 [24,25]. The mutations of the Hippo pathway result in an over-proliferative phenotype. In *Drosophila*, the core of the Hippo pathway coordinates a series of phosphorylation events that ultimately inhibit the transcriptional coactivator Yorkie (Yki). Initially, the Sterile 20-like kinase Hippo (Hpo) [26] associates with the WW domain scaffolding protein Salvador (Sav) [27] to phosphorylate and activate the DBF family kinase Warts (Wts). Subsequently, the activated Wts interacts with Mob as a tumor suppressor (Mats) [28] to phosphorylate and inhibit Yki [29], creating a 14-3-3 binding site. Yki then binds with the 14-3-3 protein, retaining Yki in the cytoplasm [30].

The Hippo pathway is evolutionarily conserved in mammals [31]. The Hippo pathway controls the expression of downstream genes, including *diap1*, *bantam* and *cyclin E* [32], to regulate proliferation and apoptosis, thereby maintaining tissue size homeostasis [33]. Furthermore, the Hippo pathway is implicated in the regulation of the lipid metabolism and organ regeneration [34]. The dysregulation of the Hippo pathway has been identified in various cancers, such as malignant mesothelioma [35] and prostate adenocarcinoma [36]. A previous study has found that YAP1 increases the glutamine level and enhances de novo nucleotide synthesis in zebrafish [37,38]. The knockdown of YAP/TAZ decreased the expression of CTPS1 in MDA-MB-231 cells [39]. However, the mechanism by which the Hippo pathway regulates nucleotide synthesis remains unknown. Therefore, our study aimed to investigate the connection between the Hippo pathway and cytoophidia.

Using the *Drosophila* egg chamber as a model [40], we observed that the Hippo pathway and cytoophidium assembly interconnect in posterior follicle cells (PFCs). Mutations in the Hippo pathway result in over-proliferation and Notch accumulation in the PFCs. Hippo pathway mutations reduce the length and number of cytoophidia within the PFCs. In addition, Hippo pathway mutations reduce the length of the cytoophidium, even when CTPS is overexpressed. CTPS knockdown suppresses the over-proliferative phenotype in *hpo* mutant PFCs. Therefore, our study connects the Hippo pathway and cytoophidia in PFCs.

## 2. Results

### 2.1. Hippo Pathway Is Essential for Oocyte Polarity

In *Drosophila* oogenesis, the establishment of the body axis depends on the polarization of the oocyte [41,42]. The asymmetric distribution of mRNA and proteins within the oocyte plays a pivotal role in determining the anterior–posterior (AP) body axis [43]. For example, *gurken* (*grk*) mRNA localizes adjacent to the oocyte nucleus [44], while *bicoid* (*bcd*) mRNA is found at the anterior pole [42], and *oskar* (*osk*) mRNA is concentrated at the posterior pole [41]. The dysregulation of the Hippo pathway disrupts oocyte migration, leading to the over-proliferation of anterior and posterior follicle cells, as well as the mis-localization of Osk and Staufen (Stau) [45,46].

We used a mosaic analysis with a repressible cell marker (MARCM) technique [47] to generate FRT/FLP mitotic clones of mutated cells with a loss of function (*wts^x1^*) allele of *wts*. In control egg chambers, the OSK signal was localized in the crescent at the posterior side (Figure 1A,B), while single-layer follicle cells surrounded the germline cells (Figure 1C). The Notch intracellular domain (NICD) signal was observed at the apical part of the follicle cells (Figure 1A,D). The GFP signal indicated that the follicle cells expressed GFP but had no mutation (Figure 1E). In the *wts^x1^* egg chambers, when the posterior follicle cells were GFP-positive clones (*wts^x1^* clones) (Figure 1F,J), we observed that the OSK mis-localized to the middle of the oocyte (Figure 1G), and we also observed the over-proliferative phenotype (Figure 1H) and the accumulation of the NICD signal (Figure 1I). These results are consistent with those of previous studies [45,46]. We found that the signal of GFP in the *wts^x1^* PFCs was lower than other GFP-positive clones (Figure 1J), but the NICD signal accumulated in the *wts*-mutant PFCs. Hence, in order to mark the *wts^x1^* PFCs, we chose the NICD signal as a marker (the yellow line in Figure 1I). The over-proliferative phenotype is effectively framed by the yellow line (Figure 1H).

### 2.2. Cytoophidium Length and Number Are Decreased in wts-Mutant PFCs

To investigate whether the *wts^x1^* mutation affects CTPS cytoophidia assembly in PFCs, we used antibodies targeting the NICD and CTPS for the immunostaining of egg chambers. In the control egg chambers, cytoophidia were observed in germline cells and follicle cells (Figure 2A,D). The NICD signal localized at the apical side of the follicle cells (Figure 2B), and a layer of follicle cells surrounded the germline cells (Figure 2C). In the *wts^x1^* egg chambers shown in Figure 2F, the accumulated NICD signal is framed by a yellow line (Figure 2G) to mark the *wts^x1^* PFCs, and the over-proliferative phenotype was observed, as shown with the yellow line (Figure 2H). We compared the length of cytoophidia in the NICD accumulation area with a normal area (Figure 2I). Comparing the GFP-positive clones, which had no accumulated NICD, with the GFP-negative clones, we found that the length of cytoophidia had no difference (data no shown) in the *wts^x1^* egg chambers. An analysis revealed that the length of cytoophidia in the accumulated NICD was shorter compared to the adjacent cells (Figure 2K). By counting the numbers of cytoophidia and follicle cells, we found that the proportion of cytoophidium in the accumulated NICD PFCs was decreased compared to other GFP-positive clones and GFP-negative clones (Figure 2L).

### 2.3. Cytoophidium Length and Number Are Decreased in Hpo-Mutant PFCs

To confirm that the reduction in the cytoophidium length can indeed be regulated by the Hippo pathway, we generated FLP/FRT mitotic clones in the egg chamber using a truncating (*hpo^BF33^*) allele of *hpo*. In the control egg chambers (Figure 3A), the NICD signal (Figure 3B) and cytoophidium length (Figure 3D) had no significant differences between the GFP-positive and GFP-negative clones (Figure 3E). There was a layer of follicle cells surrounding the germline cells (Figure 3C). The GFP-positive clones indicated that the follicle cells expressed GFP but had no mutations.

In the *hpo^BF33^* egg chambers (Figure 3F), the accumulated NICD signal is framed by the yellow line in Figure 3G. The over-proliferative phenotype was also observed in the *hpo*-mutant PFCs (Figure 3H). Furthermore, during the quantification of the cytoophidia length, we observed that the *hpo^BF33^* resulted in a decrease in the cytoophidium length in the NICD accumulation area (Figure 3I,K) compared with the cytoophidia in other follicle cells. Similarly, the proportion of cytoophidium in accumulated NICD PFCs was decreased compared to the other GFP-positive clones and GFP-negative clones (Figure 3L).

### 2.4. Hpo Mutation Decreases Cytoophidium Length Even If CTPS Is Overexpressed

To investigate whether CTPS overexpression can reduce the cytoophidia length in the *hpo* mutant PFCs, we generated a stock containing the *hpo^BF33^* mutation and *UAS-CTPS-mCherry-HA*. In the control with the CTPS overexpressed egg chamber (Figure 4A), the yellow lines shown in Figure 4A marked the GFP-positive clones, which overexpressed CTPS. The NICD signal localized at the apical side of the follicle cells (Figure 4B). The follicle cells surrounded the germline cells (Figure 4C). The length and abundance of cytoophidia increased in the GFP-positive clones (Figure 4D).

In *hpo^BF33^* with CTPS overexpressed egg chambers (Figure 4F), the *hpo* mutant PFCs shown in 4G are framed by a yellow line based on the accumulated NICD signal. The over-proliferative phenotype was observed and is shown within the yellow line in Figure 4H. The cytoophidium length increased within the GFP-positive clones (Figure 4F,I). The GFP-positive follicle cells took *hpo* mutation and overexpressed CTPS (Figure 4J). Additionally, it is worth noting that the length of cytoophidia in the accumulated NICD clones was shorter than the that in the other GFP-positive clones (Figure 4I,K).

### 2.5. CTPS Knockdown Suppresses hpo^BF33^-Induced Excessive Proliferation

To investigate whether the CTPS is required for cell proliferation regulated by the Hippo pathway, we generated a stock containing *hpo^BF33^* and *UAS-CTPS-RNAi*. Here, we chose the *hpo^BF33^* egg chamber as a control (Figure 5A), and the surface matched the accumulated NICD signal (Figure 5A’–A’’’). In *hpo^BF33^* with CTPS knockdown egg chambers (Figure 5B), we observed that the NICD signal accumulated in the *hpo^BF33^* with CTPS knockdown PFCs (Figure 5B’). By comparing the volume of surface, we found that the NICD accumulated volume in the *hpo^BF33^* with CTPS knockdown was smaller than that in the *hpo^BF33^* egg chambers (Figure 5E). This result indicated that CTPS knockdown suppresses the over-proliferation induced by *hpo^BF33^* mutation.

### 2.6. CTPS Overexpression Increases the Volume of NICD in Hpo Mutant PFCs

Although the *hpo^BF33^* mutation still resulted in a reduction in the cytoophidium length in PFCs when the CTPS was overexpressed, we attempted to determine whether CTPS overexpression exacerbates over-proliferation associated with *hpo^BF33^*. In the *hpo^BF33^* with CTPS overexpressed egg chamber (Figure 5C), the volume of NICD accumulation was observed, and it is marked in a light red surface in Figure 5C’’–C’’’. The quantification showed that CTPS overexpression increased the volume of NICD accumulated in *hpo^BF33^* with CTPS-OE PFCs by comparing with the *hpo^BF33^* egg chamber (Figure 5E).

### 2.7. CTPS^H355A^ Overexpression Does Not Increase the Volume of NICD in Hpo Mutant PFCs

To investigate whether the cytoophidium structure affects cell proliferation controlled by the Hippo pathway, we generated a stock containing *hpo^BF33^* and *UAS-CTPS^H355A^-mCherry-HA*. It is worth noting that the cytoophidium is depolymerized when the His355 is replaced by Alanine. The H355A mutation only affects polymerization, but not the enzymatic activity [22,48,49]. In *hpo^BF33^* with *CTPS^H355A^* overexpressed egg chambers (Figure 5D), the GFP-positive clones took *hpo^BF33^* mutation and overexpressed CTPS^H355A^. The quantification showed that the volume of NICD in *hpo^BF33^* with *CTPS^H355A^-OE* had no significant difference compared with the *hpo^BF33^* clones (Figure 5E). These results suggest that the cytoophidium structure contributes to the over-proliferative phenotype induced by the Hippo pathway.

## 3. Discussion

The Hippo pathway regulates cell proliferation and apoptosis. We found that mutations of *wts* and *hpo* decrease the length and number of cytoophidium in PFCs. A previous study found that the overexpression of dMyc promotes cytoophidium assembly and that the knockdown of dMyc reduces cytoophidium formation and the nucleus size in follicle cells [10]. The knockdown of dMyc downregulated CTPS expression in *Drosophila* cells [50]. The overexpression of active Ras increases the length and abundance of cytoophidia [11]. The overexpression of dMyc induces cell growth, and the overexpression of active Ras induces over-proliferation in *Drosophila* intestines. The cytoophidium length reduction phenotype induced by Hippo mutations differs from dMyc and Ras.

Cytophidium formation is related to the intracellular CTPS concentration, and the overexpression of CTPS promotes cytoophidium assembly and increases the CTP concentration [51]. The role of the Hippo pathway in the regulation of CTPS expression remains unclear. We hypothesize that the mutation of the Hippo pathway does not result in an abnormal expression of CTPS. CTPS cytoophidia inhibit enzyme activity in *S. cerevisiae*, *Escherichia coli* and *D. melanogaster* [9,51,52,53]. During stage 10a-b of oogenesis, the follicle cells grow rapidly and the number of cytoophidia is reduced [10]. Hence, we speculate that the CTPS cytoophidia may depolymerize to release more active CTPS for CTP synthesis in Hippo-mutant PFCs.

In *Drosophila* oogenesis, follicle cells are suitable for the study of cell proliferation. A previous study found that mutations in the Hippo pathway control the polarity of posterior follicle cells, and Dlg could be found all around the cell periphery in Hippo-mutant PFCs [46]. The cell polarity maintains the cytoophidia formation. The cytoophidia distribute in the basolateral side of follicle cells, and the absence of aPKC decreased the percentage of cells with cytoophidia [21]. Hence, the reduced proportion of cytoophidia in Hippo-mutant PFCs may be due to a disruption in the cell polarity.

In the future, we can further investigate whether CTPS expression is regulated by the Hippo pathway. Our study showed that over-proliferation was promoted when CTPS was overexpressed. The knockdown of CTPS suppressed the over-proliferation induced by the Hippo pathway mutation, indicating that CTPS plays a role in the regulation of cell proliferation by the Hippo pathway. Therefore, modulating the activity of metabolic enzymes may offer potential treatment options for cancer caused by Hippo mutations. Hence, it is possible to use 6-Diazo-5-oxo-L-norleucine (a CTPS inhibitor) [17,54] to target CTPS to suppress the over-proliferative phenotype induced by Hippo mutations.

In conclusion, our data demonstrate that the Hippo pathway regulates CTPS cytoophidium formation in *Drosophila* posterior follicle cells. Interestingly, when CTPS is overexpressed, the Hippo pathway still decreases the length of cytoophidia. Additionally, the knockdown of CTPS suppresses the over-proliferation caused by *hpo^BF33^*. Collectively, this study provides a novel insight into the research on Hippo-related cancer by establishing a connection between the Hippo pathway and CTPS cytoophidia.

## 4. Materials and Methods

### 4.1. Fly Stocks

All *D. melanogaster* stocks were maintained at 25 °C on standard cornmeal [55].

The MARCM system:

*hsFlp; FRT42 tubulin-GAL80, tubulin-GAL4*/Sp; *UAS-GFP*/Tm6B;

*yw, hsFlp, UAS-GFP; FRT82B tubulin-GAL80, tubulin-GAL4*/Tm6B;

*yw; FRT42D hpo^BF33^*/Cyo;

Control: *yw; FRT42D*/Cyo;

*yw; FRT82B wts^latsX1^*/Tm6B;

Control: *yw; FRT82B*/Tm6B.

The transgenic RNAi line, *UAS-CTPS-RNAi*, was obtained from the TsingHua Fly Center, stock number THU2302. The homozygote *UAS-CTPS-mCherry-HA* in Chromosome III and the homozygote *UAS-CTPS^H355A^-mCherry-HA* in Chromosome III were constructed by our lab. If/Cyo; Sb/Tm6B was a gift from *Yuu Kimata’s* lab.

We generated *Hpo^BF33^*/Cyo; *UAS-CTPS-RNAi*/Tm6B by crossing *Hpo^BF33^*/Cyo with double balancer flies for two generations to construct *Hpo^BF33^*/Cyo; Sb/Tm6B. *UAS-CTPS-RNAi* flies were crossed with double balancer flies for two generations to construct If/Cyo; *UAS-CTPS-RNAi*/Tm6B. Then, *Hpo^BF33^*/Cyo; Sb/Tm6B were crossed with If/Cyo; *UAS-CTPS-RNAi*/Tm6B to obtain the *Hpo^BF33^*/Cyo; *UAS-CTPS-RNAi*/Tm6B strain. The control strain, *FRT42D*/Cyo; *UAS-CTPS-RNAi*/Tm6B, was constructed using the same method.

We used the same method to construct the following stocks:

*Hpo^BF33^*/Cyo; *UAS-CTPS-mCherry-HA/*Tm6B*;*

*FRT42D*/Cyo; *UAS-CTPS-mCherry-HA*/Tm6B;

*Hpo^BF33^*/Cyo; *UAS-CTPS^H355A^-mCherry-HA/*Tm6B.

### 4.2. Immunohistochemistry

Female *Drosophila* were incubated for 3–5 days after pupation and dissected in at least three technical replicates. Each replicate contained more than five flies. Mosaic tissues were obtained with the FLP/FRT system with hsFLP drivers. The follicle cell clones were generated by the FRT/FLP recombinase with the presence of GFP as a marker for homozygous clones. For the loss of function, adult flies were heat-shocked for 1 h at 37 °C and heat shocked for 1 h at 37 °C at the third day, and then dissected 3 days after heat shock.

Ovaries from flies were dissected in Grace’s Insect Medium (Thermo Fisher, Rockford, IL, USA) and then fixed in 4% formaldehyde (10 μL of 40% formaldehyde (Sigma-Aldrich, Shanghai, China) diluted in 90 μL PBS) for 10 min before immunofluorescence staining.

The samples were then washed using PST (0.5% horse serum + 0.3% Triton X-100 in PBS) and incubated in PST for about 1 h to block. The samples were incubated in primary antibodies overnight at room temperature. Then, they were briefly rinsed with PST and incubated in secondary antibodies and Hoechst 33342 (1:5000; Thermo Fisher, Rockford, IL, USA) for two hours at room temperature.

Primary antibodies used in this study were rabbit anti-CTPS (1:1000; y-88, sc-134457, Santa Cruz BioTech Ltd., Santa Cruz, CA, USA), mouse anti-Notch (1:100, C17.9C6, Developmental Studies Hybridoma Bank, USA) and rabbit anti-Oskar (1:500). Secondary antibodies used in this study were goat anti-mouse Alexa Fluor^®^ 594 nm (1:500, Invitrogen A11032), donkey anti-rabbit Cy3 (1:1000, Jackson 711-165-152), donkey anti-mouse Cy5 (1:1000, Jackson 715-005-151) and donkey anti-rabbit Cy5 (1:1000, Jackson 711-175-152).

### 4.3. Microscopy and Image Analysis

All images were obtained under a laser-scanning confocal microscope (Zeiss 980). Image processing was performed using Zeiss Zen (version 3.4). For each statistical quantification, we collected the images using Zeiss 980 with the interval as 0.5 μm for z-stack by using 40× objective, and more than 3 egg chambers were quantified per genotype (biological repeats = 3). Each view was processed for maximum intensity projection, and the length of cytoophidium length was analyzed using ImageJ version 1.43 U (downloaded from https://imagej.net/ij/index.html). The number of cytoophidia and cell nuclei were counted using multi-point tool in ImageJ [56]. To quantify over-proliferative phenotype, we chose the volume of accumulated NICD signal based on NICD accumulated in Hippo-mutant PFCs. The volume of accumulated NICD was boxed by the surface tool in Imaris x64 9.0.1. Representative images with maximum intensity projection are shown in all figures.

### 4.4. Data Analysis

Images collected using confocal microscopy were processed using Adobe Illustrator, ImageJ and Imaris x64 9.0.1. Quantitative analysis was processed using Excel (Version 2108) and GraphPad Prism 9. For all experiments, the data are represented as average ± SEM. Student’s *t*-test was performed to check for significant differences between data groups. In dot plots, a dot represents an egg chamber.

## Figures and Tables

**Figure 1 ijms-25-01453-f001:**
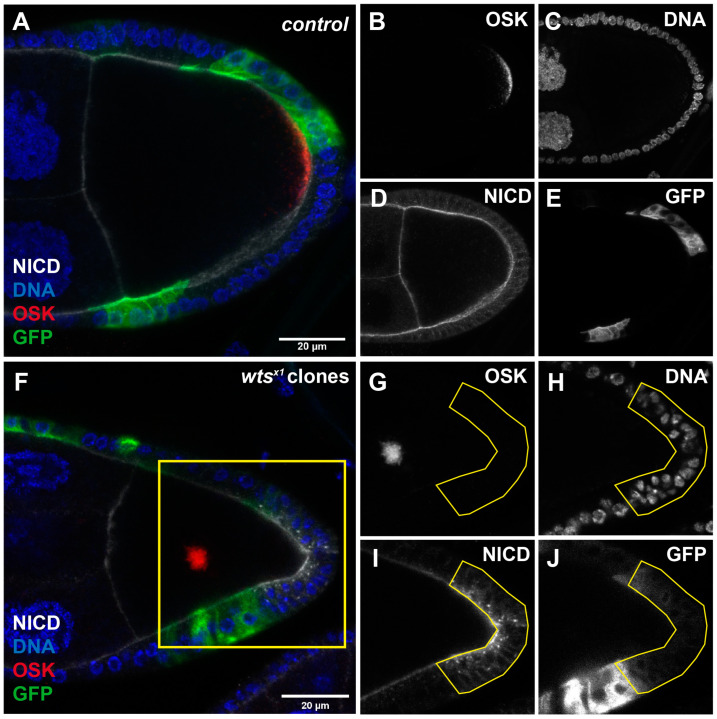
*Wts* is required for oocyte polarity. (**A**–**E**) The Oskar (OSK, red in (**A**), white in (**B**)) protein localizes at the posterior crescent of the stage-9 FRT82B control oocyte. The Notch intercellular domain (NICD, white in (**A**,**D**) labels the Notch signal and is located in the membrane of the follicle cells and germline cells. Recombinant cells are marked with GFP, and the GFP-positive clones express GFP but have no mutation (green in (**A**), white in (**E**)). (**F**–**J**) When *wts* mutates in posterior follicle cells, OSK (red in (**F**), white in (**G**)) relocates to the middle of the stage-9 oocyte. The *wts* mutant posterior follicle cells show an over-proliferative phenotype and an accumulation of NICD (white in (**F**,**I**)). DNA is stained with Hoechst 33342 (blue in (**F**), white in (**H**)). Recombinant cells are marked with GFP, and GFP-positive clones take *wts^x1^* mutation (green in (**F**), white in (**J**)). Scale bars, 20 μm.

**Figure 2 ijms-25-01453-f002:**
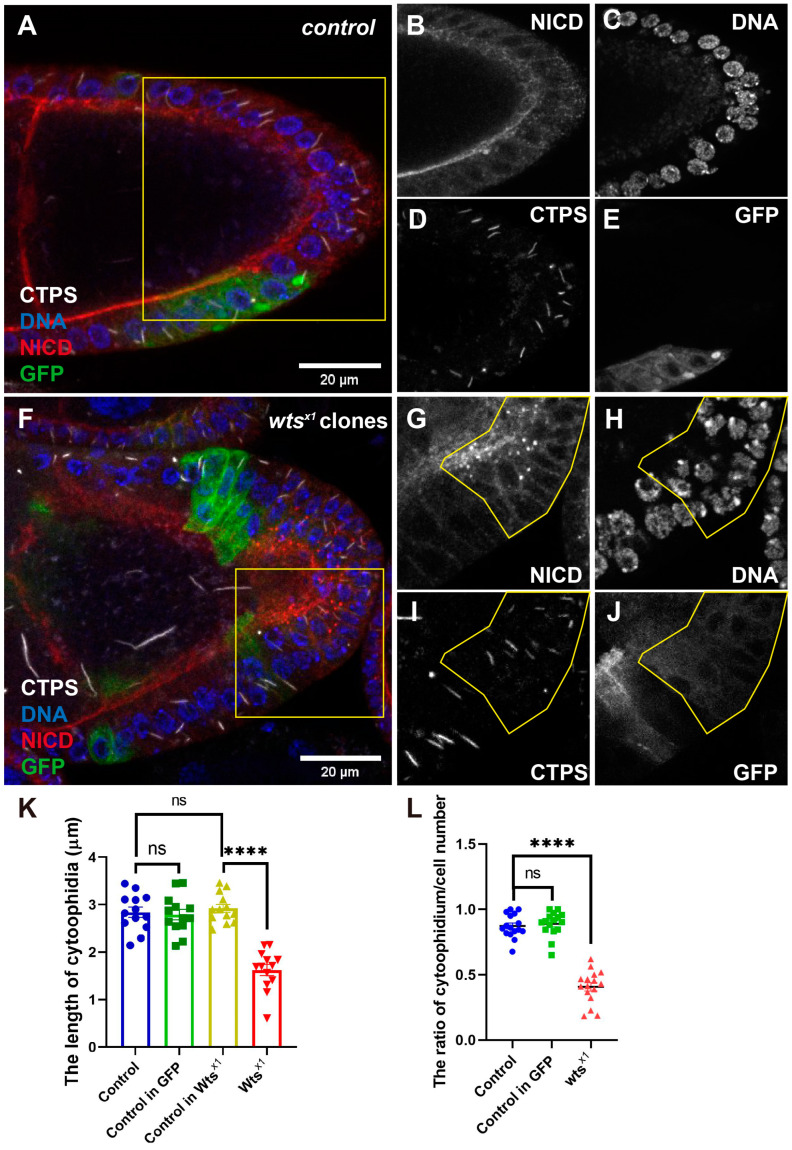
*Wts^x1^* decreases cytoophidium length and number in *wts* mutant PFCs. (**A**–**E**) In the FRT82B control egg chamber, there is no difference in the NICD signal (red in (**A**), white in (**B**)) or the nuclear size (blue in (**A**), white in (**C**)) of follicle cells between GFP-positive (green in (**A**), white in (**E**)) and GFP-negative clones. Cytoophidia labeled with CTPS (white in (**A**,**D**)) exist in follicle cells and germline cells. There is no difference in cytoophidium length between GFP-positive and GFP-negative clones. Recombinant cells are marked with GFP and have no mutation. (**F**–**J**) In *wts^x1^* egg chamber, NICD (red in (**F**), white in (**G**)) accumulates in posterior follicle cells (yellow line in (**G**)). After *wts^x1^* mutation, the length of cytoophidia (white in (**F**,**I**)) in posterior follicle cells is decreased, while the number of cells increases. DNA is stained with Hoechst 33342 (blue in (**F**), white in (**H**)). Recombinant cells are marked with GFP (green in (**F**), white in (**J**)) and take *wts^x1^* mutation. Scale bars, 20 μm. (**K**) Quantification shows that *wts^x1^* decreases the cytoophidium length in posterior follicle cells. **** *p* < 0.0001, ns = not significant, mean shown with SED (Student’s *t* test). (**L**) Quantification shows that *wts^x1^* decreases the number of cytoophidium in posterior follicle cells. **** *p* < 0.0001, ns = not significant, mean shown with SED (Student’s *t*-test).

**Figure 3 ijms-25-01453-f003:**
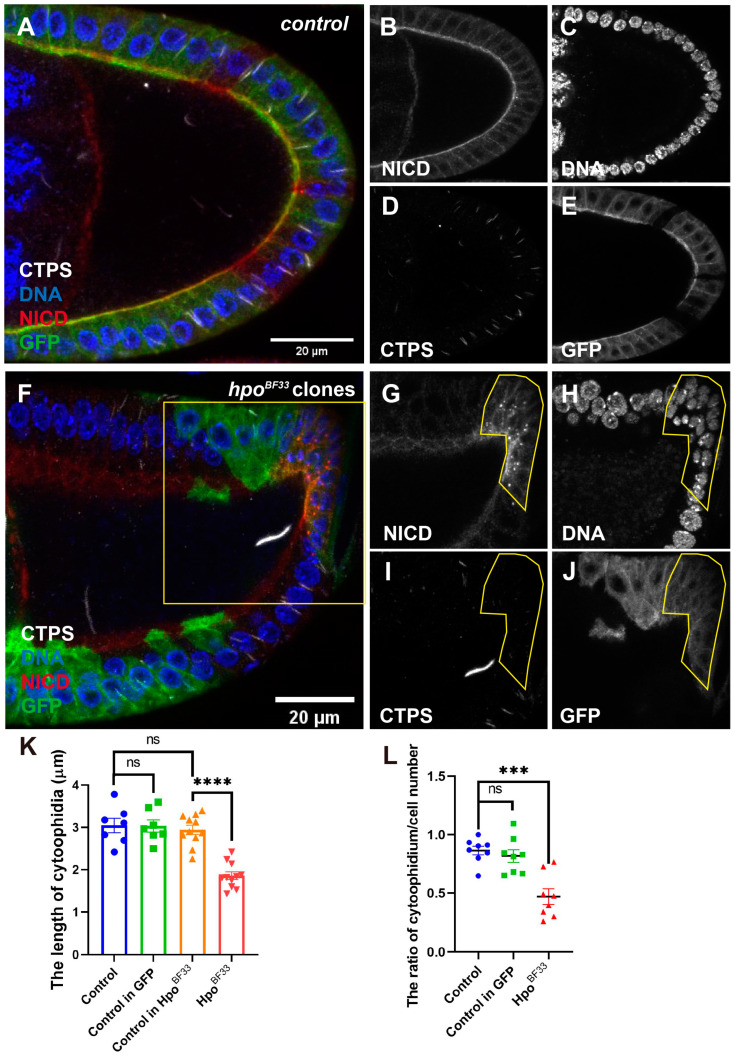
*Hpo^BF33^* decreases cytoophidium length and number in *hpo* mutant PFCs. (**A**–**E**) In the FRT42D control egg chamber, there is no difference in the NICD signal (red in (**A**), white in (**B**)) and the nuclear size (blue in (**A**), white in (**C**)) of follicle cells between GFP-positive and GFP-negative clones. Cytoophidia labeled with CTPS (white in (**A**,**D**)) exist in follicle cells and germline cells. There is no difference in cytoophidium length between GFP-positive and GFP-negative clones. Recombinant cells are marked with GFP and have no mutation (green in (**A**), white in (**E**)). (**F**–**J**) In the *hpo^BF33^* egg chamber, NICD (red in (**F**), white in (**G**)) accumulates in posterior follicle cells (yellow line in (**G**)). After *hpo^BF33^* mutation, the length of cytoophidia in posterior follicle cells is decreased, while the number of cells increases. DNA is stained with Hoechst 33342 (blue in (**F**), white in (**H**)). Recombinant cells are marked with GFP (green in (**F**), white in (**J**)) and take *hpo^BF33^* mutation. Scale bars, 20 μm. (**K**) Quantification shows that the *hpo^BF33^* decreases the cytoophidium length in posterior follicle cells. **** *p* < 0.0001, ns = not significant, mean shown with SED (Student’s *t* test). (**L**) Quantification shows that *hpo^BF33^* decreases the number of cytoophidium in posterior follicle cells. *** *p* < 0.001, ns = not significant, mean shown with SED (Student’s *t*-test).

**Figure 4 ijms-25-01453-f004:**
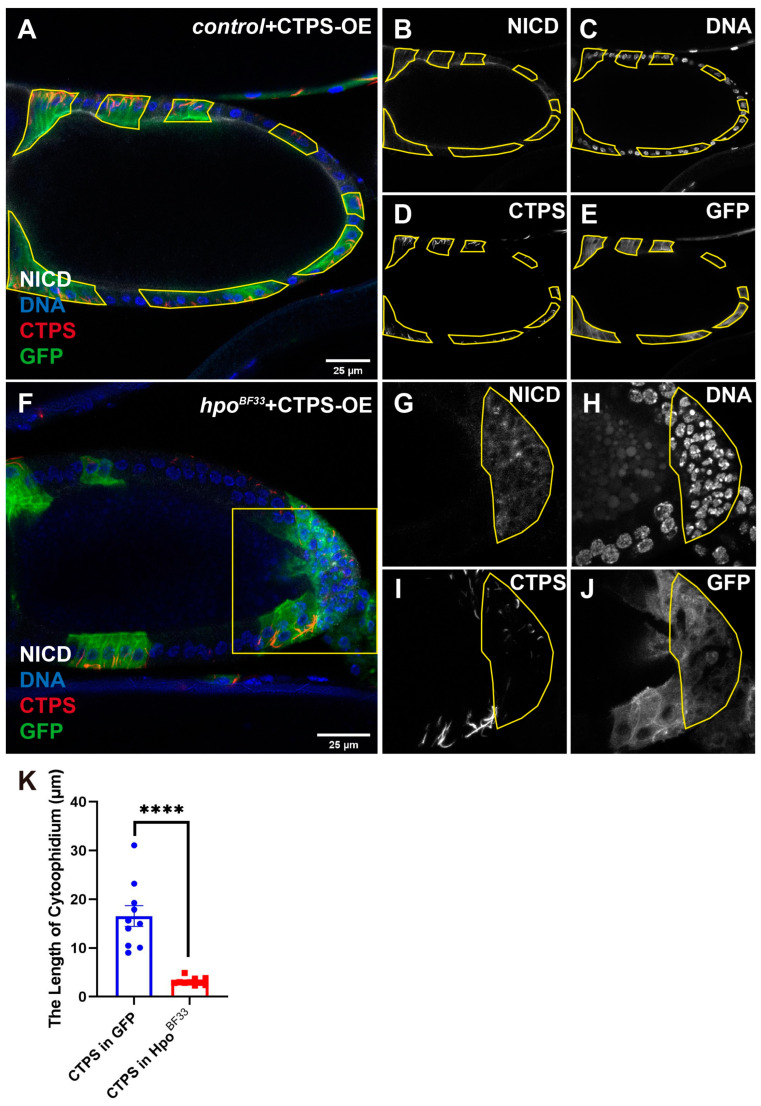
Even if CTPS is overexpressed, the *hpo^BF33^* decreases cytoophidium length in PFCs. (**A**–**E**) In the FRT42D with CTPS overexpression egg chamber, there is no difference in the NICD signal (white in (**A**,**B**)) of follicle cells between GFP-positive (green in (**A**), white in I) and GFP-negative clones. The length and number of cytoophidia labeled with CTPS (red in (**A**), white in (**D**)) increase in the GFP-positive follicle cells. DNA is stained with Hoechst 33342 (blue in (**A**), white in (**C**)). The GFP-positive clones are marked by yellow line. (**F**–**J**) In the *hpo* mutant with CTPS overexpression, NICD (white in (**F**,**G**)) accumulates in posterior follicle cells (yellow box in (**F**)). After *hpo^BF33^* mutated, the length of cytoophidium in posterior follicle cells (yellow box in (**I**)) is shorter than those in other GFP-positive regions. DNA is stained with Hoechst 33342 (blue in (F), white in (**H**)). Recombinant cells are marked with GFP (green in (**F**), white in (**J**)) and take *hpo^BF33^* mutation and CTPS overexpression. Scale bars, 25 μm. (**K**) Quantification shows that *hpo^BF33^* mutation decreased the cytoophidium length in posterior follicle cells, even if CTPS was overexpressed. **** *p* < 0.0001, mean shown with SED (Student’s *t*-test).

**Figure 5 ijms-25-01453-f005:**
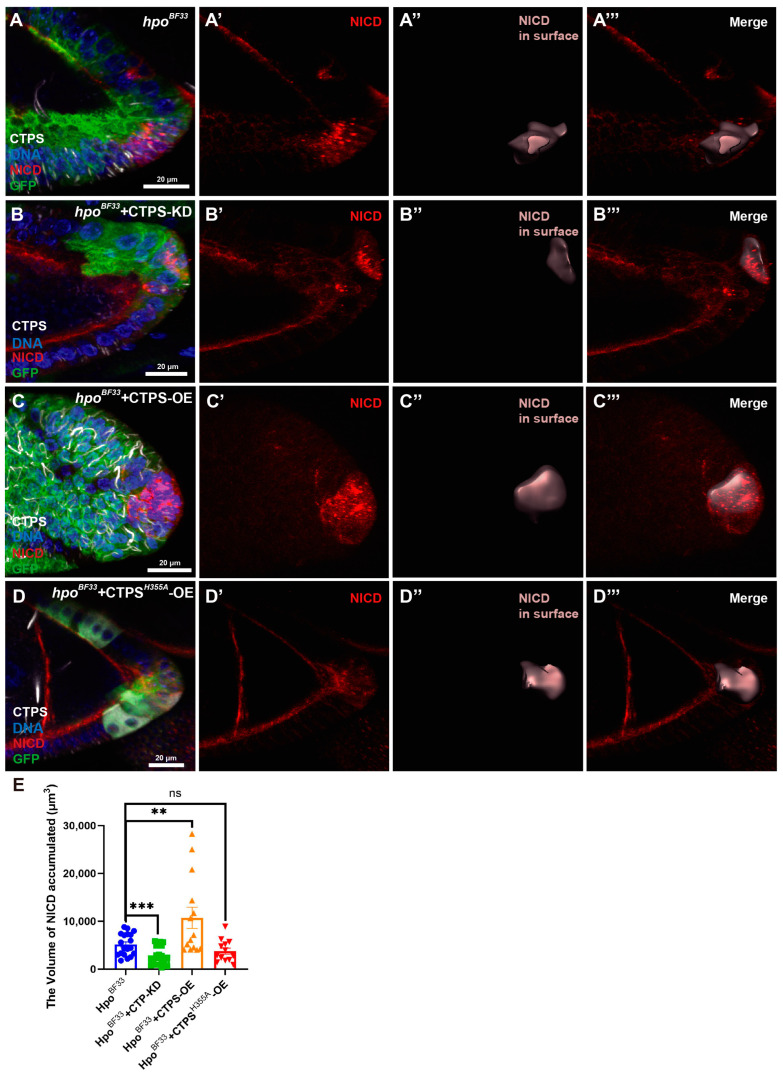
CTPS plays a role in the over-proliferative phenotype in *hpo^BF33^* mutant PFCs. (**A**–**A’’’**) In the 3D view of *hpo^BF33^* mutant egg chamber, NICD (red in (**A**,**A’**)) accumulates in posterior follicle cells (light red box in (**A’’**,**A’’’**)). The merged image shows the surface within the accumulated NICD signal. (**B**–**B’’’**) In the 3D view of *hpo^BF33^* with CTPS knockdown, NICD (red in (**B**,**B’**)) accumulates in posterior follicle cells (light red in (**B’’**,**B’’’**)). The merged image shows the surface within the accumulated NICD signal. (**C**–**C’’’**) In the 3D view of *hpo^BF33^* with CTPS overexpression, NICD (red in (**C**,**C’**)) accumulates in posterior follicle cells (light red in (**C’’**,**C’’’**)). The merged image shows the surface within the accumulated NICD signal. (**D**–**D’’’**) In the 3D view of *hpo^BF33^* with *CTPS^H355A^* overexpression, NICD (red in (**D**,**D’**)) accumulates in posterior follicle cells (light red in (**D’’**,**D’’’**)). The merged image shows the surface within the accumulated NICD signal. Scale bars, 20 μm. (**E**) Quantification shows that CTPS overexpression promotes the volume of over-proliferative phenotype in *hpo^BF33^* PFCs. CTPS knockdown inhibits the volume of over-proliferative phenotype in *hpo^BF33^* PFCs. CTPS^H355A^ overexpression has no difference with *hpo^BF33^* egg chambers in the volume of accumulated NICD signal. ** *p* < 0.01, *** *p* < 0.001, ns = not significant, mean shown with SED (Student’s *t*-test).

## Data Availability

All data is available in the main text.

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
