# Peer review of "Connecting Hippo Pathway and Cytoophidia in Drosophila Posterior Follicle Cells"

_ijms, 2024, doi:10.3390/ijms25031453_

Round 1
Reviewer 1 Report
Comments and Suggestions for Authors
The paper of Weng et al. on hippo pathway and cytoophidia using the posterior follicle cells of Drosophila is an interesting study. However, it makes for a difficult read in some instances due to the English language used in the paper. The authors should improve upon the level of English language to make reading easy. Scientific names in the paper should also be stated in full at first mention.
The tense used in the abstract should be in the past. The authors should also indicate the importance of the study (and a justification for the study of a linkage between the hippo pathway and CTPS/cytoophidia) since it does not come out clearly in the abstract.
The authors kept stating the aim of the specific activities with their justifications in the Results. These should be deleted since they best fit the Materials and Methods.
The specific concentrations and/or volumes of reagents and other chemicals used in the study should be provided.
The age and specific species of Drosophila used in the study should clearly be stated.
In the Discussion, several parts (indicated in the reviewed paper) are literature review and not discussion. The discussion should be improved by comparing your findings to other studies indicating similarities and drawing contrast where applicable while proffering reasons for the differences and your observations/findings.
Other comments have been made in the reviewed paper.

Comments on the Quality of English LanguageThe quality of English language needs improvement.
Reviewer 2 Report
Comments and Suggestions for Authors
with genetic approach, and by looking at PFC, authors found the genetic interaction between CTPS and hippo pathway.
Specific points:
1, it's better move Fig 1 to supplementary fig, as the phenotype has beed described before.
2, it will be great if author use reporter gene "bantam, diap1", as readout to assess the interaction.
3, author shall do H3S10p, to see the changing of cell proliferation of PFC.
last, authors shall discuss more why hyper proliferation of PFC carrying less CTPS or reduced size
Reviewer 3 Report
Comments and Suggestions for Authors
Comments on the Quality of English LanguageIn general, the English is sufficent to be understood.
Round 2
Reviewer 3 Report
Comments and Suggestions for Authors
In general, the authors' answers are satisfied, except for comment #2 and #5.
2. There appears to be an inconsistency in the designation of GFP-negative cells as mutated in Fig. 1F and 2F, which contrasts with the representation in Fig. 3F and 4F. Could the authors clarify this discrepancy?
Response: We apologize for the confusion caused by our results. Based on previous studies, the Hippo pathway had been shown to affect the proliferation of posterior follicle cells in Drosophila. However, the previous studies used GFP-negative as the mutation marker (Polesello, C.; Tapon, N., Salvador-warts-hippo signaling promotes Drosophila posterior follicle cell maturation downstream of notch. Curr Biol 2007, 17, (21), 1864-70.). In our study, we used GFP-positive signal as the marker. In Hippo mutant posterior follicle cells, the GFP signal becomes weaker, which makes the region we outlined with a yellow line appear similar to GFP-negative clones. In the line 100-103, we pointed out that there was a weakening of GFP signal in wts mutant PFCs and we chose the NICD accumulation signal as a marker.
Comment: In my assessment, the current explanation regarding the yellow line, which appears distinctly black in my observation, remains unconvincing. It would be advantageous for the authors to present more compelling and clearer results to support their claims. Additionally, I do not see the description about the weak GFP expression in line 100-103.
5. Could the authors describe the means of the circles in Fig. 4A, 4B, 4C, 4D, and 4E?
Response: Thanks to the reviewer for the question. The Fig 4A-E is the control with CTPS-OE egg chamber. The yellow line marked the GFP-positive clones and framed the region of CTPS overexpression. In this region, the number and abundance of cytoophidia are increased.
Comment: Please add this description in the manuscript.
